# Towards sustainable land management in small islands: A Water-Energy-Food nexus approach

Romain Authier[1]*, Benjamin Pillot[1], Guillaume Guimbretière[2]¤, Pablo Corral-Broto[3], Carmen Gervet[1]

**1** ESPACE-DEV, Univ Montpellier, IRD, Univ Guyane, Univ Réunion, Univ Perpignan, Montpellier, France, **2** LACY, CNRS, Univ Réunion, Saint-Denis, France, **3** ESPACE-DEV, Univ Réunion, IRD, Univ Montpellier, Univ Guyane, Univ Perpignan, Saint-Pierre, France

¤ Current address: Now at TREE, CNRS, Université de Pau et Pays de l'Adour, Pyrénées-Atlantiques, France
* romain.authier@ird.fr

**Data Availability Statement:** You can find the Python code and data used as input of the model using the following link: https://github.com/romain-authier/Land_Use_Competition_Small_Islands.git.

## Abstract

The environmental and multi-sectoral challenges faced by small islands requires consideration of sustainability issues. The sustainability challenges in these regions involve in particular the achievement of a greater autonomy through the development of local resources. This is a complex system that encompasses interconnections between the resources available and the land use. In this article we focus on the study of the Water-Energy-Food (WEF) nexus, and propose an integrated and systemic approach to do so. Our contribution consists in studying food system sustainability of small islands by exploring the reciprocal influences between the valorization of local WEF resources and land use competition for various integrated WEF scenarios. Additionally, we integrate dietary behaviors and demonstrate their close interlinking with land use practices, and thus their impact on the potential for transitioning towards a more sustainable food system. To achieve this, we present a generic combined Geographical Information Systems (GIS) and robust optimization model. This model is then applied to Reunion island using collected real data. Our approach aims to assist local policymakers, at the island scale, by constructing insightful scenarios to facilitate informed decision-making. Our results highlight the need to save land space when developing local resources through effective land use management policies combined with a shift in food practices. This shift would imply in particular, to convert some of the sugarcane areas into subsistence farming. Furthermore, the results emphasize the importance of transitioning consumption practices under various integrated WEF scenarios, showcasing our model as an insightful decision-support tool.

## Introduction and related work

Small islands can be defined as territories surrounded by water, with a small surface area, limited resources and characterized by strong economic, climatic and demographic vulnerabilities [1]. More precisely, *small islands* are characterized by important environmental issues such as

**Funding:** PhD grant from IRD (Institut de Recherche pour le Développement). The funders had no role in study design, data collection and analysis, decision to publish, or preparation of the manuscript.

**Competing interests:** The authors have declared that no competing interests exist.

the land fragility or biodiversity loss, notably because of their geographical isolation, their exposure to natural risks and to anthropic pressure [2]. In addition, these islands present a great vulnerability to climate change, as well as energy vulnerabilities, through their dependence on fossil fuel imports, leading to greater exposure to any economic disruption such as rising energy prices or supply shortages [3], food vulnerabilities with high dependence on food imports [4] and land vulnerabilities due to biophysical, socio-economic and demographic factors [5]. Indeed, due to strong territorial constraints, land use is a major issue for these islands where the preservation of natural spaces is essential, especially when they experience population growth [6]. All of these vulnerabilities, specific to the island context, require a *sustainability* goal especially since *small islands* have strong territorial constraints and a geographical isolation. Sustainability evaluation criteria can be tested within these territories before being scaled up to a broader context [7]. Sustainability for small islands is a complex concept to define, lacking consensus, and strongly dependent on the geographic and demographic context [8], but also on the historical context often marked by a colonial heritage [9]. However, several ideas revolve around this concept, including the decision-making paradigm shift, governance and community involvement, and finally, resilience [7]. The IPCC defines this idea of resilience as follows: *Resilience is defined [. . .] as the ability of a social-ecological system and its components to anticipate, reduce, accommodate, or recover from the effects of a hazardous event or trend in a timely and efficient manner* [10]. The resilience of these socio-ecological systems represented by *small islands* is a key aspect in the search for sustainability [11], and a lack of resilience implies an increased vulnerability of these territories which are already fragile and highly dependent on international trade (imports and exports) [12]. Indeed, the vulnerability of these islands to external shocks such as the Covid crisis and the associated border closures [13], the impact of climate change on international trade [14], or the impact of international conflicts on exports further increases the need for resilience in these territories. This objective of resilience implies improving water management [15] and considering greater autonomy harnessing local resources [16] by the development of renewable energies and local food systems to support energy self-sufficiency process [17] and food self-sufficiency process [18]. Nevertheless, land use is a core issue to tackle the challenges of multiple resource management and planning, especially if one aims to maximize local resources use in a highly spatially constrained territory [8]. A suitable and promising approach to address the multiple and interdependent challenges is the integrated Water-Energy-Food (WEF) nexus. This nexus approach explores the connections between water, energy and food resources to better understand their interdependence in the context of sustainable development. There is a growing interest in studying the equilibrium of food, energy, and water resources within the WEF nexus, especially under various exogenous constraints including climate change, population growth, and urban growth [19].

**On a world scale**, the study conducted by [20] analyses the various issues surrounding WEF resources management. It reviews waste management and identifies crisis surrounding these resources, as well as the main solutions for dealing with declining resource availability and accessibility. This study provides insightful information on the future sustainability of these resources. On the same scale, the study presented in [21] proposes to analyze the impacts of different agricultural production and food consumption scenarios on WEF resources production and associated environmental consequences using a combination of two Integrated Assessment Models (IAM): IMAGE 3.0 (Integrated Model to Assess the Global Environment) and FDM for food and demand production. This study identifies the different impacts of the scenarios on water withdrawal, energy consumption, crop and grass consumption, as well as the land-use impact in relation to agricultural production. Another modelling approach proposed by [22] consists in an optimal planning of the WEF nexus through stochastic

optimization modelling under constraints using GAMS (General Algebraic Modeling System) optimization software. This decision-support model is an extension of the model proposed by [23] in deterministic form. Optimization consists in minimizing economic costs (electricity production costs, fixed and variable costs associated with power plants, water consumption costs, food production costs) and environmental costs (greenhouse gases emission costs) considering a 15-year planning horizon. Results present an optimal electricity production for each type of fuel (here coal and natural gas), as well as specific considerations regarding water use (ground or surface water) and optimal food production.

**On a national scale**, [24] propose a systemic approach to study food security issues in South Africa. This study highlights the reciprocal influences of energy and water resource prices on food prices, as well as the role of water and energy systems on food quality, availability and affordability. It provides a set of practical solutions for improving food security, taking into account its interactions with other resources. A spatio-temporal decision support model was developed by [25] for the assessment of energy requirements in the agricultural sector in Saudi Arabia. A web-based Geographical Information Systems (GIS) application was created to facilitate user selection of a site on a map. Then, from various input agricultural crop information, the corresponding water, land and energy requirements are returned to the user. To achieve this, an agriculture land suitability map is created based on several factors. The tool developed is intended as a decision-support tool, providing a global view of the interactions between food, energy and water at a crop level and at several spatial points in order to spare the use of these resources.

**On a local scale**, methodological approaches are varied and are presented as decision-support tools. It is the case of [26] through a multi-objective optimization model for the evaluation of securing WEF resources according to sustainability, availability and security criteria in the Monterrey Metropolitan Area (in Mexico). The optimization consists in minimizing, successively, economic costs related to the nexus, freshwater consumption and greenhouse gases emissions for the design of energy and water distribution networks. In this way, the securing of resources varies based on the objective function, enabling stakeholders to find the best compromises. Another decision-support model presented by [27] consists in a mixed-integer non-linear modelling applied to Yucheng Station (China) where land use is fully integrated into the model. The aim is to find the best combinations of agricultural crops based on different objectives such as maximizing profit and agricultural yields, and minimizing water, energy consumption and $CO_2$ emissions. In this way, land use competition between different agricultural crops is illustrated and the study reports on optimal planning for crop rotation.

**On an island scale**, [28] provide a decision support-tool through scenarios production to analyze future trends in energy consumption and greenhouse gases emissions on the island of Tenerife (Spain), based on projections of demand for water and energy resources, as well as demographic, economic and climatic variables. Two different scenarios are proposed for 2050, a maintained trend scenario in which renewable energies are underdeveloped, and an ecology aware scenario with an electricity mix made up of 100% renewable energies. This study highlights the island's dependence on imports as a function of the increase in demand for water and energy according to the scenarios. The approach developed by [29] encompasses all three dimensions of the WEF nexus. The concept of sustainability for the Carribean Small Islands Developing Sates (SIDS) is analyzed using several indicators selected to assess the sustainability of each island concerning the food, water and energy components. The analysis incorporates multiple data sets, including climate and population growth projections for 2050. This study provides a holistic view of the concept of sustainability and highlights specific features of each Carribean SIDS. The risks associated with the nexus (in terms of balance between demand and supply) due to demographic growth and economic development have been highlighted by

[30]. For this purpose, an indicator quantifying the balance between demand and supply was built. The study consists in a material flow analysis to assess the risks associated with the nexus on Kinmen island. Results show that a high risk is linked to high resources consumption by industries and households, giving us important information on the impact of population growth and industrial development on securing WEF resources in islands. [31] constructed a framework and developed a user-friendly nexus platform based on GIS in Taiwan: *GREAT for FEW*. This tool is based on a life cycle assessment approach to analyze the links between WEF resources. It enables environmental impacts to be quantified for different time horizons on the basis of parameters linked to local crop production and energy production. The integration of land use data as input enables the generation of various maps as impacts distribution maps or land use allocation map as output. A key aspect of the tool is the preservation of agricultural lands for food security, and when this objective is achieved, the remaining agricultural lands are converted into industrial/urban lands to support population and demographic growth. This study enables to investigate the influences of food security on land-use change dynamics, and to study the trade-offs between bioenergy production, food supply and environmental benefits. Finally, [32] proposed a spatio-temporal modelling of drivers of change that influence food and energy self-sufficiency through semi-directive interviews in Reunion island (French overseas department). Land use maps are generated following different land use scenarios (extrapolation, food+, bioelectricity+, planning+) for 2035. This article gives a clear vision of the trends associated with food and energy self-sufficiency process as a function of time, and of the issues surrounding land availability in small islands, seen as a resource limiting the development of local food production and energy projects. Land use competition between urban growth, food and energy is also evaluated.

**In summary**, many approaches tackle parts of the problem we want to address, with different focus of studies and at different scales. On a global scale, WEF nexus studies give a systemic view of the issues associated with the nexus and the factors to be considered to enhance sustainability from both economic and environmental perspectives. However, the approaches developed cannot be directly applied to the small island scale, as these studies overlook the spatial dimension and land use constraints linked to the WEF nexus. On a national and local scale, some studies incorporate spatial data and territorial specificities. Nevertheless, there is a lack of exploration into territorial land-use dynamics related to land use competition. On an island scale, existing literature does not address sustainability in terms of land use competition induced by the use of local WEF resources.

In this paper, food system sustainability in small islands is discussed by studying the reciprocal influences between the use of local WEF resources and land use competition under various integrated WEF scenarios. The corresponding research question is the following: Does the pursuit of resilience necessarily contributes to enhancing the sustainability of small islands within a systemic approach?

The key challenges we identified to assess food system sustainability in small islands through scenarios are: (1) the integration of multi-source data through projections data (WEF resources demand, urban growth) for a future horizon 2035, spatial data that enables the characterization of specific features of the territory, and qualitative data for shaping scenarios, (2) the definition and specification of heterogeneous spatio-temporal constraints reflecting the WEF nexus, (3) the construction of integrated WEF scenarios that integrate diverse food consumption profiles and various means of agricultural production to accurately characterize the food system and (4) a comprehensive model aimed at identifying the thresholds to food self-sufficiency process beyond which it affects food system sustainability. In the context of systemic modelling, the objective is to formulate a comprehensive methodology integrating data collection, constraint specification, and optimization modules to generate insightful scenarios.

The primary objective is to extract potential alternative land usages, to be determined trough the optimization process, for each cadastral parcels by establishing geographical constraints that the parcels must meet using a set of geographical data layers. The optimization process necessitates the incorporation of contextual data digitalization, including factors like population and resource demand as input. For each scenario, it allows to derive future land uses for each parcel over a 10-year projection (by 2035) while satisfying spatio-temporal constraints characterizing the WEF nexus. In this paper, we present our integrated GIS and robust optimization methodology as a comprehensive decision-support tool. This combined approach enables the modelling of interactions between WEF resources self-sufficiency process and land use competition. Thus, it becomes possible to identify thresholds to food self-sufficiency process beyond which it affects food system sustainability under various integrated WEF scenarios.

*Summary of contributions.* Our contributions can be summarized as follows: 1) Food system sustainability in small islands is addressed by examining the reciprocal influences between WEF resources self-sufficiency process and land use competition. This is accomplished through the integration of a GIS and optimization model that integrates diverse data and nexus-specific constraints, aiming to maximize a food Self-Sufficiency Ratio (SSR). 2) The development of integrated WEF scenarios is based on various consumption patterns and agricultural pathways on the island, to be incorporated into the comprehensive model. These scenarios are shaped by qualitative data collected from field interviews and existing literature.

The article is structured as follows: the methodology section provides a comprehensive overview of our methodology and outlines the inputs of the integrated model, including each of its component: the land use potential allocation model and the land use scenarios optimization model. In the case study section, the described methodology is applied to the specific case of the food system in Reunion Island, a French overseas. It shows how our model can provide a framework for land-use pathways policies that respect local resource limits for various integrated WEF scenarios. This integrated model is thought as a simple decision-support tool for policymakers aiming to enhance the resilience of the food system through a systemic approach.

## Our methodology

This paper aims to address food system sustainability regarding land use dynamics resulting from the valorization of local resources within a WEF nexus approach. One key aspect of our approach involves quantifying each dimension of the nexus in terms of its direct impact on land use. Fig 1 illustrates the conceptual model designed to provide insights into sustainability challenges within small island food systems and constitutes one of our contributions. In this study, demographic growth and economic development are considered as the main drivers influencing change within the WEF nexus [28, 33]. Urban growth, resulting from economic development and population growth, impacts not only land use through urban sprawl but also the demand for WEF resources, assuming a constant average resource consumption per inhabitant. It leads to an increased supply of these resources to meet the growing demand. The means of production implemented to meet demand will determine the island's capacity to achieve food system sustainability. In a situation where no environmental objectives are considered, increased demand for WEF resources leads to a greater reliance on imports, resulting in an unsustainable food system [34]. Conversely, in the quest for resilience, dependence on imports is reduced in favor of local WEF resources. However, the valorization of local WEF resources can strengthen land use competition, already reinforced by urban sprawl. This land use competition may also limit the development of local resources (mutual influences) due to

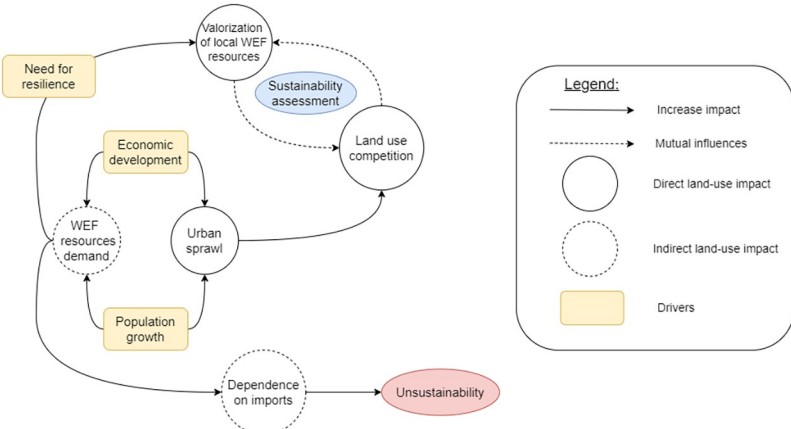

**Fig 1. Conceptual model.**

strong spatial constraints characteristic of small islands. Thus, there may be a maximum threshold for the valorization of WEF resources due to land use competition. As a result, food system sustainability in small islands is discussed through the study of reciprocal influences between WEF resources self-sufficiency process and land use competition for several integrated WEF scenarios.

From a methodological perspective, identifying thresholds to discuss food system sustainability involves addressing the following problem: How to determine an optimal land use for each parcel (1) within the framework of the WEF nexus so that (2) constraints related to the nexus are satisfied (3) while maximizing a food SSR for a future horizon 2035, and (4) under various integrated WEF scenarios?

To emphasize our main contributions, the integrated model must fulfill several objectives: (1) the consideration of land use competition in the context of local WEF resource use, and (2) the identification of thresholds to food self-sufficiency process beyond which it affects food system sustainability. To tackle the management of spatio-temporal data, the introduction of constraints linked to the nexus, and an objective function related to the food self-sufficiency process, we present a GIS-based model combined with a robust optimization model. The model itself is developed in [35] and we recall the associated workflow in Fig 2. In summary, first the land use potential allocation model takes as input various geographical data layers and enables the allocation of land use potential(s) for each parcel based on the values taken by the constraints specified upstream (land use, water requirements, slope, altitude, neighborhood and surface area). It generates a land use potentials map, which is then converted through a pre-processing step into a set of parcel attribute lists (surfaces, crop yields. . .). These lists serve as input for the land use optimization model, along with temporal data. Finally, the optimization model derives an optimal land uses map.

## Land use potential allocation

The initial step in the allocation of land use potentials involves (1) partitioning the territory into cadastral parcels and (2) specifying land use constraints to allocate land use potentials for each parcel (refer to Fig 2). Note that the land use potential for a parcel refers to its capacity to transition from its current land use to another. This concept embodies the evolution of land use within the framework of the WEF nexus.

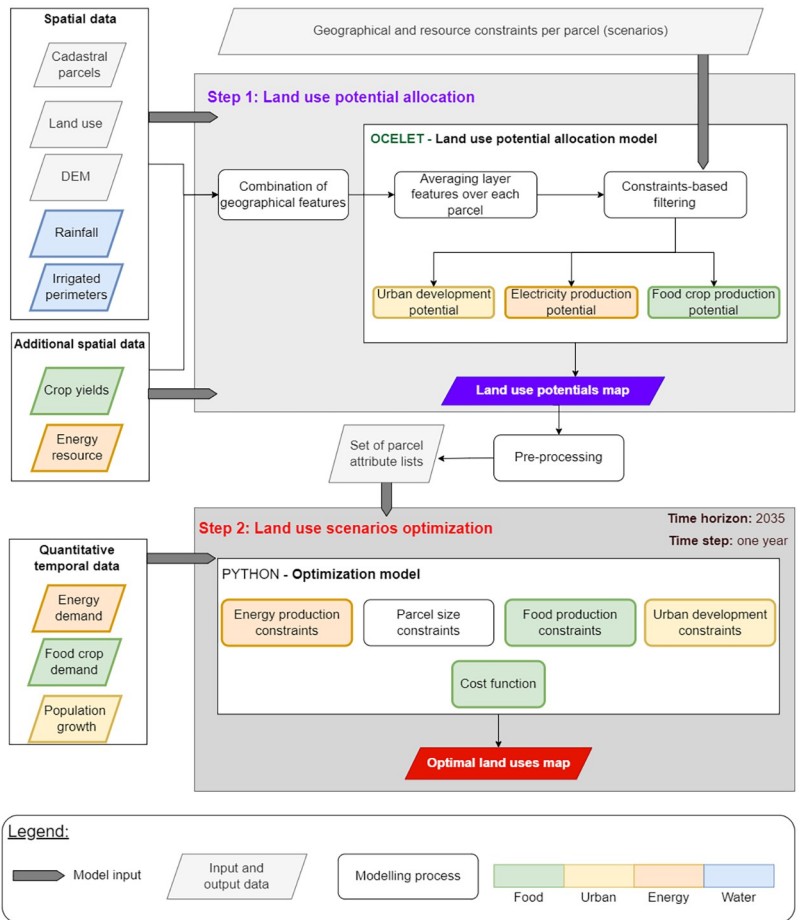

**Fig 2. Integrated methodological framework.**

**Specifications.** One key aspect of our approach involves quantifying each dimension of the nexus in terms of its direct impact on land use. In this context, direct land use impact pertains to the land footprint associated with a process (such as urban growth) or a resource (such as energy and food) and its utilization (for electricity production) within the territory.

- *Water:* The future direct land-use impact of water facilities is assumed to be negligible (improvement of the distribution network efficiency, reusing waste water... [36]). Thus, the land-use impact of water will be analyzed at a *crop-level* by considering the development of crop production in irrigated areas or within areas with sufficient rainfall to meet the crop water requirements.

- *Energy:* Talking about energy refer, here, to electricity production. The future direct land-use impact of some energy sources and future energy production projects, responding to the global rise in energy demand, is considered for local energy resources such as biomass and low-power energy density source (such as intermittent sources of energy).

- *Food:* The future direct land-use impact of food, responding to the global rise in food demand and the establishment of new dietary behaviors (through scenarios), is marked by the expansion of crop cultivation in new areas.

- *Urban growth:* The future direct land-use impact of urban growth, driven by population growth and economic development, involves the creation of new residential housing and infrastructure.

Hence, three distinct land use potentials are defined: (i) the potential for crop production, (ii) the potential for urban growth, and (iii) the potential for electricity production. The nexus approach developed here implies that some parcels have the potential to transition from their current land use to another or to multiple land uses, taking into account concomitant land uses. That is the case for agricultural parcels, which have the flexibility to either remain agricultural or change their land use to become urban or energy-producing parcels. In contrast, some parcels do not have this flexibility. This applies to natural and protected parcels, as well as urban parcels, where land use remains unchanged, although urban parcels can transition to also become energy-producing parcels through solar self-consumption.

**Land use potential allocation model.** Once all the land use potentials have been defined, the next phase involves allocating these potentials to the parcels. To achieve this, the domain-specific language *Ocelet* [37], a powerful tool for processing land-use dynamics using interaction graphs, is employed. The process of allocating land use potential must adhere to predetermined constraints. Consequently, for assigning urban growth potential, candidate parcels need to conform to defined maximum slope and distance thresholds from urban parcels, along with specific land use criteria. For the allocation of electricity production potential, candidate parcels must adhere to constraints such as minimal surface area (in $m^2$), maximum altitude (in $m$), maximum slope (in%) and specific land use. Furthermore, an additional constraint on water requirements (in *mm/year*) is considered for the allocation of crop production potential. Referring to this last constraint, the annual crop water requirements are satisfied if the parcel receives a sufficient amount of rainfall to sustain crop production or if it is located within irrigated perimeters. Note that all the constraint values are obtained from literature and field data. At the output, the model returns a map depicting land use potentials, which is then converted into a set of parcel attribute lists that subsequently serves as input for the optimization model.

**Land use optimization.** As illustrated in Fig 2, the set of parcel attribute lists input the optimization model, along with data on projected population growth, electricity and food crop demand by 2035 which are derived from estimates provided by various institutes and companies. The optimization model is an original model detailed more specifically in [35]. It is coded in Python (library used: pyomo; solver: CPLEX). For each time step, the model systematically identifies the optimal land use or combination of land uses for each parcel from its range of potential land uses, allowing for concomitant land uses on a single parcel. This process is executed under constraints associated with the WEF nexus and incorporates an objective function for the time horizon 2035, aiming to maximize the food SSR, expressed in *kcal*. The term "food SSR" in this context refers to the specific SSR for the crops considered in the modelling and is expressed in Eq 1:

$$\sum_{crops} \frac{Local\ production\ of\ crop\ c}{Total\ demand\ of\ crop\ c} * 100 \tag{1}$$

The uncertainties associated with the modelling are addressed through a robust approach, which involves employing deterministic intervals to enclose uncertainties within robust extreme values (ex: $Value = [\underline{Value}, \overline{Value}]$) as described in [38, 39]. In our reliable computing approach, we employ best-case and worst-case scenarios to explore the extremes within the solution space. The worst-case scenario represents the minimum theoretical food production, maximum food and electricity demand, and maximum urban sprawl surface based on high

population growth projections. Conversely, the best-case scenario is defined by maximum theoretical food production, minimum food and electricity demand, minimum urban sprawl surface, and is based on low population growth projections. By employing these extreme scenarios, we establish a range within which all other solutions are encompassed. This range is bounded by the components of the best-case and worst-case scenarios, ensuring the reliability and robustness of our optimization results. The optimization model returns an optimal land uses map for each time step for a specific integrated WEF scenario. The nexus approach developed involves defining multiple optimization constraints that effectively integrate the drivers influencing change within the WEF nexus, with a specific focus on the objective of resilience to enhance sustainability. Specifications of the optimization problem are given in Table 1 and details of the equations are provided in Table 2. The following constraints are delineated:

*Food production constraint.* The primary set of constraints aims to prevent excess crop production by ensuring that annual crop production is less than the projected annual demand for both the best-case scenario (lowest projected demand) and the worst-case scenario (highest projected demand) at each time step.

**Table 1. Optimization problem specifications.**

| Given: | |
|---|---|
| Unit: Year $t \in T = \{2023\ldots2035\}$ | |
| Unit: Parcel $p \in P = \{set\ of\ parcels\}$ | |
| Unit: Crop $c \in C = \{set\ of\ crops\}$ | |
| Unit: Energy source $e \in E = \{set\ of\ primary\ energy\ sources\}$ | |
| Set of parcels with a potential of production of crop $c$ | $P_c$ |
| Set of parcels with a potential of production of electricity from source $e$ | $P_e$ |
| Set of parcels with urban growth potential | $P_u$ |
| Total electricity demand for a year $t$ (*GWh*) | $d_t^{elec} = [\underline{d_t^{elec}}, \overline{d_t^{elec}}]$ |
| Existing electricity production (*GWh*) | $P^{elec}$ |
| Surface of parcel $p$ (*ha*) | $S_p$ |
| Number of households for a year $t$ | $h_t = [\underline{h_t}, \overline{h_t}]$ |
| Urban extension area per new household for a year $t$ (*ha/household*) | $Su_t^h$ |
| Surface energy density from a primary source $e$ for a parcel $p$ (*GWh/ha*) | $p_p^e$ |
| Total demand for crop $c$ for a year $t$ (*ton*) | $d_t^c = [\underline{d_t^c}, \overline{d_t^c}]$ |
| Calories per kg of crop $c$ (*kcal/ton*) | $kcal_c$ |
| Yield of crop $c$ for a parcel $p$ (*ton/ha*) | $Yield_p^c$ |
| **Find:** | |
| The optimal land use(s) for a parcel among all its potential land uses | |
| The surface areas allocated to urban growth, food and electricity production | |
| **Objective function:** | |
| Maximize food self-sufficiency ratio | |
| **Such that the following constraints hold:** | |
| Local food crop production is less than or equal to the food crop demand | |
| Total annual electricity production meets demand | |
| Fossil fuels imports are decreasing | |
| Limit on intermittent Renewable Energies (RE) existing and new production | |
| Limit on urban sprawl | |

**Table 2. Mathematical expressions of optimization variables, constraints and objective function.**

| Decision variables | (1) $\forall t \in T, \forall p \in P_u, s_{t,p} \in [0, S_p]$ |
|---|---|
| | (2) $\forall t \in T, \forall p \in P_c, s_{t,p} \in [0, S_p]$ |
| | (3) $\forall t \in T, \forall p \in P_e, s_{t,p} \in [0, S_p]$ |
| *Food production constraint.* | Best case scenario: $\forall t \in T, \forall c \in C, \sum_{p \in P_c} s_{t,p} * Yield_p^c \leq \underline{d_t^c}$ |
| | Worst case scenario: $\forall t \in T, \forall c \in C, \sum_{p \in P_c} s_{t,p} * Yield_p^c \leq \overline{d_t^c}$ |
| *Electricity production constraint.* | Best case scenario: $\forall t \in T, \sum_{e \in E} \sum_{p \in P_e} s_{t,p} * p_p^e + P^{elec} = \underline{d_t^{elec}}$ |
| | Worst case scenario: $\forall t \in T, \sum_{e \in E} \sum_{p \in P_e} s_{t,p} * p_p^e + P^{elec} = \overline{d_t^{elec}}$ |
| *Urban sprawl constraints.* | Best case scenario: $\forall t \in T, \sum_{p \in P_u} s_{t,p} = Su_t^h * (\underline{h_t} - h_0)$ |
| | Worst case scenario: $\forall t \in T, \sum_{p \in P_u} s_{t,p} = Su_t^h * (\overline{h_t} - h_0)$ |
| *Maximize food SSR.* | $\sum_{c \in C} \sum_{p \in P_c} \dfrac{s_{t,p} * Yield_p^c * kcal_c}{\sum_{c \in C} d_t^c * kcal_c} * 100$ |

*Electricity production constraint.* The second set of constraints aims to maintain a balance between electricity demand and production for both the best-case scenario (lowest projected demand) and the worst-case scenario (highest projected demand) at each time step.

*Energy imports constraint.* This set of constraints reflects the intention to cease the use of fossil fuels and maximize the use of local energy resources.

*Intermittent RE production constraint.* It involves establishing a threshold for intermittent Renewable Energies (RE) production due to the variability of these energy sources, which can impact power grid stability [40].

*Solar self-consumption constraint.* It is assumed that PV panels are installed on the roofs of some urban parcels. This constraint underscores the importance of developing local resources while minimizing the impact on land use as much as possible, allowing industrial and domestic consumers to directly consume self-generated electricity.

*Urban sprawl constraint.* At each time step, the urban sprawl area delineates urban growth resulting from expected population growth for both the best-case scenario (lowest population projections) and the worst-case scenario (highest population projections). The impact of economic development on urban growth is not assessed in this study.

*Land use conversion of potential areas.* To prevent the conversion of all potential areas into effective production areas from the initial time steps, we define a maximum surface area, contingent on the type of food crop, for the conversion of potential food crop production areas into effective food crop production areas.

## Case study: Food system sustainability in Reunion

Our generic methodological approach is now applied to Reunion island, an overseas French department located in the Indian Ocean, around 200 km southwest of Mauritius and 900 km east of Madagascar. Reunion island is a relevant study case to explore the challenges linked to

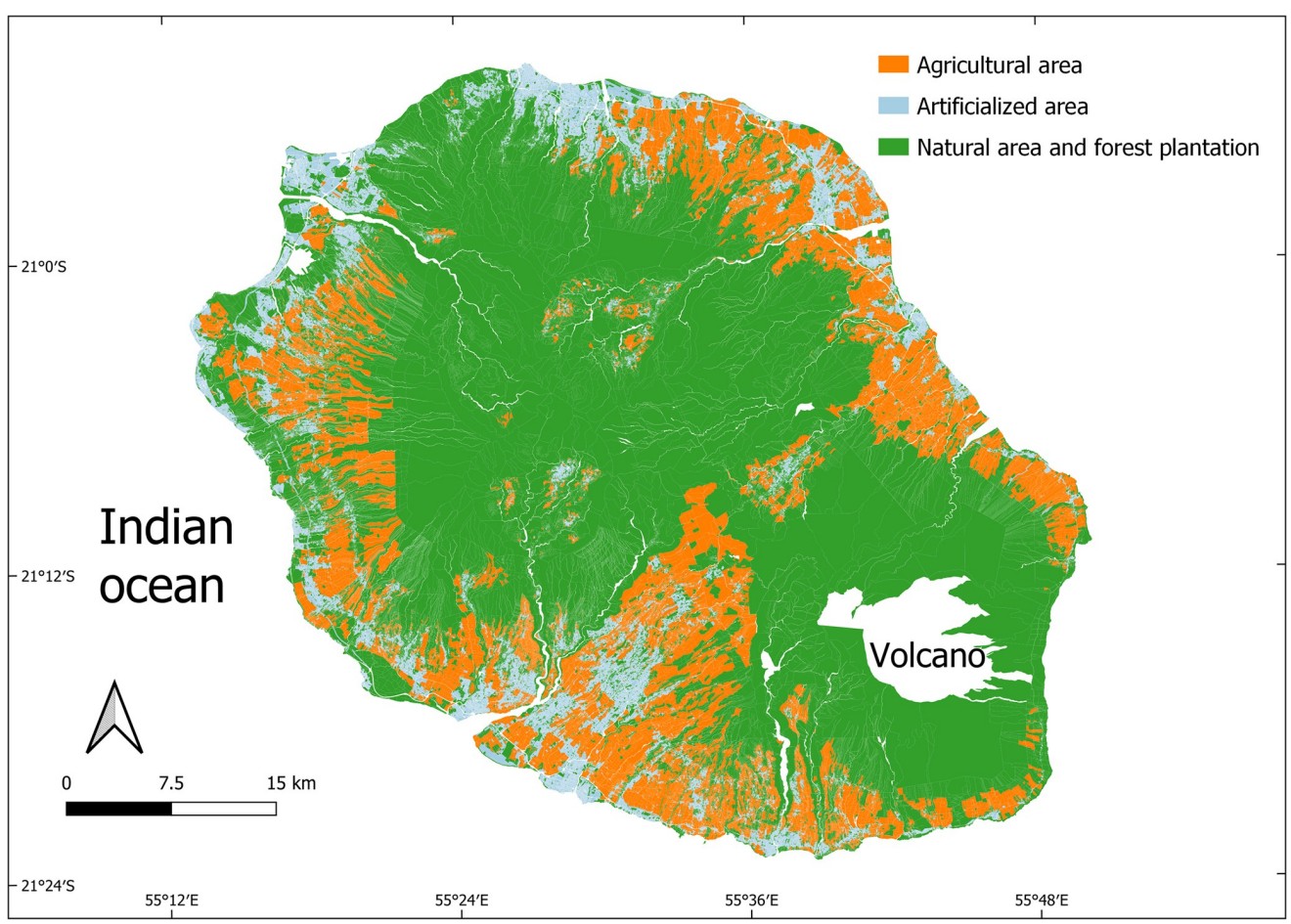

**Fig 3. Land use map delineated by cadastral parcels.**

food system sustainability in small islands to the extent that the island is highly reliant on imported food and energy resources and is vulnerable to any disruptions that could interrupt its external supply. Additionally, it faces land pressure associated with urban sprawl [41]. In this context of vulnerability, the French government expressed its desire to achieve food and energy self-sufficiency by 2030 [42]. A land-use map of Reunion island, delineated by cadastral parcels, is represented in Fig 3. It has been produced using the QGIS software based on several data layers, including a land use layer and a cadastral parcel layer (refer to the next subsection for data sources). Three main areas are highlighted: agricultural areas, artificial areas, and natural areas and forest plantations. Spaces shown in white on the map refer to spaces between parcels, such as roads or waterways, as well as the volcano. Spatial constraints are visible by the extent of protected natural areas and forest plantations, representing more than 42% of the territory. These areas are mainly located in the centre of the island while artificial and urban areas are more scattered around the coast.

## Geographical data collection

The primary data used in this study are geographical data layers (in raster and ESRI shapefile format) including land use layer, cadastral parcels layer, topography layer, irrigated perimeters

layer, practical photovoltaic power potential layer and rainfall layer. The land use layer for 2021 was obtained from CIRAD. The detailed methodology of land use classification can be found in [43]. The irrigated perimeters layer is a product of the Reunion department. The topography map is extracted from [44], and the cadastral parcel layer is sourced from [45]. Yearly data of practical photovoltaic power potential (expressed in $kWh/m^2$) are extracted from [46]. Monthly rainfall data (in $mm$) were provided by *Meteo France*. Crop yields were estimated based on a yield map for the island's agricultural sectors made by [47]. All the data were processed using QGIS software.

## WEF resources selection

Implementing a WEF nexus approach in Reunion begins with the selection of WEF resources according to the territory's specificities.

*Energy.* Various energy sources are taken into account, encompassing all sources involved in electricity production [48]. Imported energy sources include imported biomass, oil and coal, while local energy sources comprise wind, PV, hydro and local biomass. The distribution for 2022 was: oil (43%), hydro (21%), coal (19%), sugarcane bagasse (6%), imported biomass (2%) and other renewables (9%). Hereafter, local biomass will be considered to come solely from sugarcane [49]. Imported biomass, in this context, refers to wood pellets imported from North America.

*Food.* The modelling does not integrate all the food items consumed within the territory. Instead, we assume that a limited but well-selected set of food items is adequate as input to assess the overall food system sustainability within the framework of a WEF nexus approach in the sense that the approach developed here focused on studying thresholds. The chosen food items are those exhibiting the most uneven consumption patterns across the territory, based on household incomes [50], and are part of the local Creole diet: fruits, vegetables and rice.

## Land use potential assessment

The direct land use impact of WEF resources and urban growth has been evaluated to observe land use change dynamics and land use competition.

*Energy.* A distinction is made here between the direct land use impact of the energy resource and energy production systems from a specific resource. Regarding the *energy resource*, the direct land use impact linked to biomass from agricultural residues (in this case, sugarcane bagasse) is taken into account. The direct land use impact of imported resources (coal, oil, imported biomass) is zero, as well as other local resources (wind, solar, water flows). Regarding *energy production systems*, in accordance with the objectives outlined and the strategy for the development of energy projects, ground-mounted solar PV systems contribute to land use competition due to their low power density [49]. Concerning on-shore wind turbines, the challenge in the coming years is to renew the wind farms. Therefore, future production is not regarded as a contributing factor in the competition for land use. Future electricity production projects involving coal and oil are not considered in the modelling, so no direct land use impact is expected. Prospects for hydroelectricity depend more on optimizing existing power plants, so the land use impact of future hydroelectric power generation projects is considered negligible. Finally, the direct land use impact of imported biomass is also considered negligible.

*Food.* The future direct land-use impact associated with rice, fruits and vegetables is taken into account.

*Water.* The future direct land use impact of water resource and facilities is considered negligible [36].

*Urban growth.* The future direct land use impact of urban growth is considered as a consequence of population growth on the island is the extension of the urban plot (urban growth in non-urbanized areas).

## Integrated WEF scenarios

One of the main contribution of this article is the construction of integrated WEF scenarios, combining both demand-side and supply-side approaches, thanks to online literature and field interviews. Beyond simple scenarios characterizing the orientation of consumer demand (*demand-side approach*) [51], the integration of the means of production (*supply-side approach*) [52] enables a more comprehensive description of the nexus. Indeed, *demand-side approaches* can be challenging to target and direct toward the means of production. These approaches can therefore be seen as complementary (rather than replacing) *supply-side approaches*.

**Contribution of field interviews.** Over a period of four months in Reunion, numerous interviews were conducted with a diverse range of local stakeholders involved in the nexus. These interviews enabled us to collect non-public data from the department but one of the primary objectives was to contribute to scenario development. Consequently, scenarios were shaped through the gathering of both qualitative and quantitative information derived from these interviews. From these interviews, it became evident that there is a growing awareness of the vulnerabilities of the island, particularly its dependence on food imports. This awareness has given rise to associations such as *Association Riz Réunion* and association *Riziculteurs Péi 974*, whose goal is to reintroduce rice cultivation on the island (the main foodstuff consumed locally). Additionally, citizen groups such as *Oasis Réunion* position themselves as sources of proposals to public authorities, aiming for complete food self-sufficiency. While the number of stakeholders is increasing, there are still challenges in communication among them due to their divergent future paths and approaches. To these stakeholders, a range of questions were asked, including: *"What do you understand by food self-sufficiency process, and what is your opinion on achieving 100% food self-sufficiency by 2030 (governmental objective)? What socio-cultural levers do you identify to achieve greater food self-sufficiency process? What is the role of organic agriculture in the agricultural landscape of Reunion Island?"* The answers collected have helped refine the construction of the scenarios. The interviews also raised the issue of sugarcane cultivation that is subject of much debate. Some view the island's main crop as an obstacle to the goal of achieving food self-sufficiency. Meanwhile, others advocate for the preservation of this crop, putting forward a variety of arguments, particularly economic ones. Specifically, we were unable to contact *Tereos*, an agro-industrial actor in the cane-sugar-rum-energy value chain, to discuss matters related to food self-sufficiency and the diversification of sugarcane. Sugarcane is therefore a major pillar of the food system, and the impact of revisiting the levels of production is assessed through scenarios. Finally, an interview with the economist Gaëlle Pothin regarding her work on food consumption patterns in Reunion [53] guided us to differentiate several food consumption profiles in order to broaden the spectrum of the Reunionese food system, illustrating the impact of food consumption patterns on food system sustainability.

The produced scenarios are therefore dedicated to portraying the diversity of visions among local stakeholders and propose compromises to facilitate decision-making.

**Scenarios specifications: Demand-side.** *Water.* Several climate factors can influence crops water requirements including temperature, humidity or even wind speed. To simplify, crops water requirements per ha are assumed to remain constant over time.

*Energy.* Existing consumption data from 2022 are considered as the baseline to draw projections data for electricity demand for the horizon 2035 [54]. Electricity demand depends on various factors, including economic development, household growth, and changing lifestyles [49]. However, annual differences in electricity consumption across the island are partly attributed to population growth [54]. For simplicity's sake, we're assuming the same standard of living for all inhabitants and constant electricity consumption by large consumers, local authorities and business customers. Thus, the future increase in electricity consumption will only depend on population growth (number of households on the island) and the expansion of the electric vehicle fleet [49].

*Food.* Similarly, we have considered existing consumption data from 2022 as the baseline to draw projections data for food demand for the horizon 2035. The amount of a specific food consumed on the island is assumed to vary over time according to (1) the food consumption profile and (2) population growth. Then, for a specific food consumption profile, the increase in food demand will only depend on population growth according to INSEE's low population projections (best case scenario) and high (worst case scenario) [41]. In the case of Reunion, two food consumption profiles are selected according to current food practices on the island [50]. Here, we consider that both profiles are oriented towards local production when the food item considered is produced locally.

- *Traditional profile:* High consumption of rice and poultry, coupled with a low intake of fruits and vegetables. The *Traditional* profile refers here to the Creole profile.

- *Mediterranean profile:* Lower consumption of rice and poultry and a higher consumption of fruits and vegetables compared to the *Traditional* profile.

**Scenarios specifications: Supply-side.** *Water.* Crop production takes place in areas with either irrigation systems or sufficient rainfall to meet crop water requirements.

*Energy.* Targets for reducing electricity generation from fossil fuels have been formulated by [49]. On this basis, it is assumed that imports of fossil fuels, such as oil and coal, will decline. Hence, energy production will be mainly ensured by local resources (solar, wind, hydro and local biomass) and imported biomass to face the decrease in fossil fuels supply. Electricity production is reduced to zero for coal and by the end of 2030 for oil.

*Food.* In Reunion, 54% of the Utilized Agricultural Area (UAA) is devoted to sugarcane cultivation. Fodder crops account for 28% of the UAA. The remaining 18% is devoted to fruits and vegetables, and traditional sectors such as aromatic plants, coffee, vines and lentils [55]. Nevertheless, in the context of building resilience, the challenge is to increase food production for local population, necessitating a shift in agricultural practices toward subsistence farming. Thus, two sets of future agricultural trajectories are presented:

- *Subsistence farming:* agriculture intended for local population is preserved to the detriment of sugarcane, which is the primary crop for exports. It illustrates the commitment to support food self-sufficiency process through the promotion of subsistence farming.

- *Current agricultural practices:* current surface areas dedicated to each crop are maintained, constraining future crop production (rice, fruits, vegetables), ground-mounted PV production and urban growth in agricultural wastelands or in rotation with vegetable crops in the case of rice cultivation *(Association Riz Réunion)*.

From this point, various scenarios are designed by combining different food consumption profiles (demand-side) with agricultural practices (supply-side) to create four distinct integrated scenarios as depicted in Fig 4.

**Fig 4. Integrated WEF scenarios.**

The specifications of constraints for each land use potential are summarized in Table 3 for scenarios S1, S2, S3 and S4. Note that we consider only one production cycle per year for rice crop, lasting four months between November and February. Some constraints implemented into the optimization model (refer to Fig 2) need to be specified in the case of Reunion and are summarized in Table 4. Data related to the static parameters of the optimization model are summarized in Table 5.

## Results and analysis

The presented results support decision-making aimed at enhancing food system sustainability for the time horizon 2035 by addressing (1) the identification of thresholds to food self-sufficiency process for all integrated WEF scenarios and (2) the impact of scenarios on land use share. These results provide policymakers with various criteria for assessing food system sustainability using integrated scenarios and a systemic approach. The aim of this decision-support tool is to enable decision-makers to make informed decisions on land use management and to promote incentives for shifting food consumption practices.

**Scenarios influence on thresholds identification.** One key aspect of guiding decision-making involves identifying thresholds to food self-sufficiency process based on various integrated WEF scenarios. As outlined in the conference paper [35], the composition of the electricity mix can influence the food SSR (as defined in Eq 1), especially when energy projects

**Table 3. Constraints specifications for land use potential allocation per parcel in the Ocelet model.**

| Constraints | Rice | Urban | PV | Vegetables and fruits |
|---|---|---|---|---|
| Land use type (S2, S4) | Agricultural wastelands, vegetable crops, sugarcane | Agricultural wastelands | Agricultural wastelands, sugarcane | Agricultural wastelands, sugarcane |
| Land use type (S1, S3) | Agricultural wastelands, vegetable crops | Agricultural wastelands | Agricultural wastelands | Agricultural wastelands |
| Minimal surface area | 240 $m^2$ (Association Riz Réunion) | / | 3000 $m^2$ (author) | / |
| Maximal altitude | 1200 m (Association Riz Réunion) | / | / | 1800 m [43] |
| Maximal slope | 10% [57] | 30% [58] | 10% [59] | 30% [60] |
| Minimum water requirements | 300 $mm/cycle$ (Association Riz Réunion) | / | / | 300 $mm/year$ [60] |
| Neighborhood distance with urban parcels | / | 20 m (author) | / | / |

**Table 4. Specifications of constraints for optimization.**

| Optimization constraints | Specifications |
|---|---|
| Wind power | Electricity production from wind power is set to increase due to the renovation project of the Sainte Suzanne wind farm, with an expected production of 50 GWh/year from 2023. Additionally, there is an annual production of approximately 2.3 GWh from the Sainte Rose wind farm (data from 2021). Therefore, we assume a maximum annual electricity production of 53 GWh [61] for the time horizon 2035. |
| Hydropower | Electricity production from hydro is strongly correlated with the level of precipitation. Then, to simplify, electricity production from hydro is considered to be stochastic, following a uniform distribution within a specified range, which is determined based on hydropower data spanning from 2000 to 2021 [61]. |
| Local biomass | Electricity production thanks to local biomass depends on the variable amount of sugarcane bagasse collected over time [61]. For simplicity, we assume here that the quantity of bagasse harvested depends on sugarcane production areas and sugarcane yields, which are extracted from Russeil's yield map [47]. |
| Land use conversion of potential areas | Maximum surface area of conversion assessed by the author due to absence of existing data. |

with significant land use impacts, such as ground-mounted PV installations, are being developed. We observe a maximum 4.5% reduction in food SSR between an electricity mix reliant on imported biomass and another characterized by a high share of ground-mounted PV. In this article, we select an electricity mix that relies on ground-mounted PV with limited biomass imports (noting that fossil fuels imports are set to 0 for 2030 [49]). This choice has a less favorable impact on food SSR due to increased land use competition but mitigates dependence on energy resource imports.

**Table 5. Static optimization parameters.**

| Parameters | Values | References |
|---|---|---|
| Electricity production from hydropower in 2022 (GWh) | 634.2 | [48] |
| Electricity production from oil in 2022 (GWh) | 1327 | [48] |
| Electricity production from coal in 2022 (GWh) | 581.1 | [48] |
| Electricity production from PV in 2022 (GWh) | 266.3 | [48] |
| Electricity production from wind in 2022 (GWh) | 3.489 | [48] |
| Electricity production from local biomass in 2022 (GWh) | 181.4 | [48] |
| Electricity production from imported biomass in 2022 (GWh) | 50.6 | [48] |
| Domestic electricity production in 2022 (GWh) | 1313 | [54] |
| Non domestic electricity production in 2022 (GWh) | 1507 | [54] |
| Rice calories (kcal/kg) | 2800 | [62] |
| Vegetables calories (kcal/kg) | 248.6 | [55] |
| Fruits calories (kcal/kg) | 570.9 | [55] |
| *Trad profile*—Rice consumption (kg/household) | 123 | [50] |
| *Trad profile*—Fruits consumption (kg/household) | 88.4 | [50] |
| *Trad profile*—Vegetables consumption (kg/household) | 125.5 | [50] |
| *Med profile*—Rice consumption (kg/household) | 67.8 | [50] |
| *Med profile*—Fruits consumption (kg/household) | 252.7 | [50] |
| *Med profile*—Vegetables consumption (kg/household) | 263.4 | [50] |
| Urban extension area per new household ($m^2$/household) | 247 | [63] |
| Ratio of tons of bagasse per ton of sugarcane (%) | 0.30 | [61] |
| Ratio of electricity production per ton of bagasse (GWh/t) | 0.00047 | [61] |
| Rice yield (ton/ha) | [2.8, 3.3] | *Association Riz Réunion* |

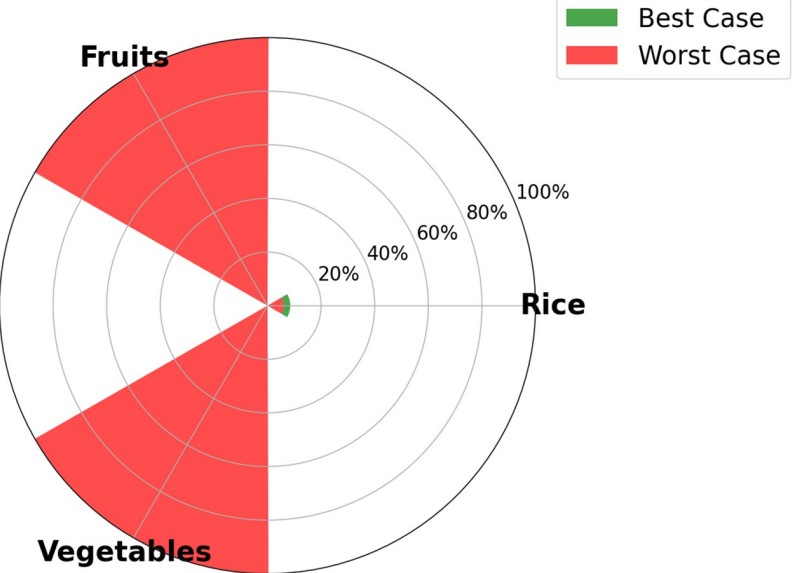

**Fig 5. Specific crop SSR for scenario S1 (BAU).**

One of the output of the optimization model is the visualization, for each time step, of the food SSR for each crop considered in the model. The maximum food SSR specified for each crop by 2035 is depicted in Fig 5 for scenario S1, in Fig 6 for scenario S2, Fig 7 for scenario S3, and Fig 8 for scenario S4 (refer to Fig 4 for scenarios specifications). These results are visually presented through a donut chart with gauges that indicate the levels of food SSR. The donut chart effectively encapsulates the graphical representation of the results, highlighting both the best-case and worst-case scenarios within its circular design.

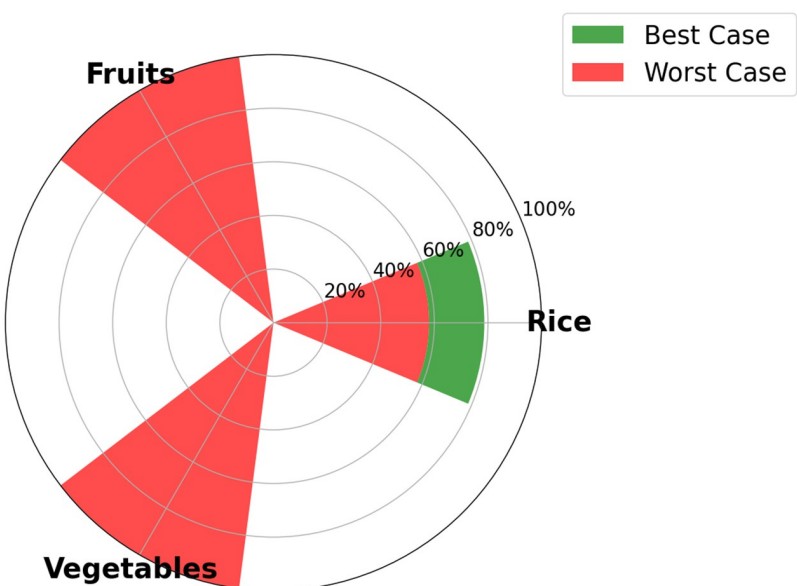

**Fig 6. Specific crop SSR for scenario S2 (Agricultural shift allowed).**

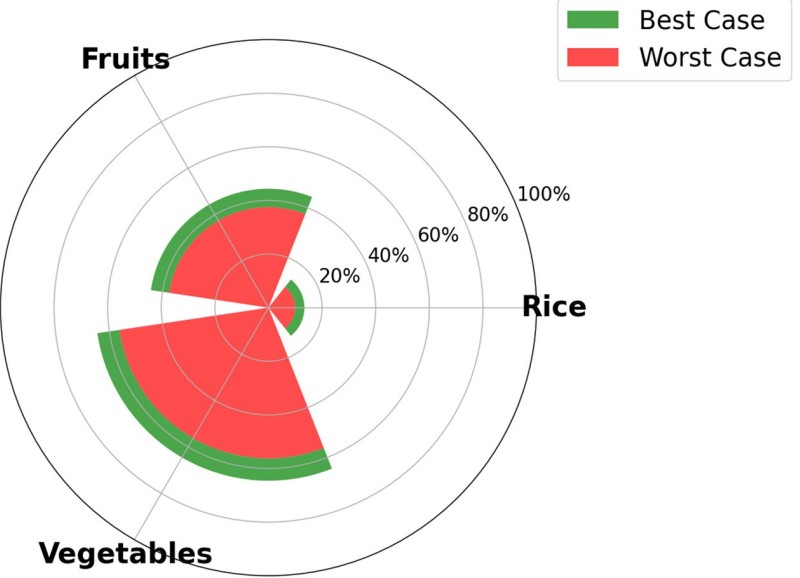

**Fig 7. Specific crop SSR for scenario S3 (Dietary shift).**

At first sight, we note that the food SSR reaches 100% for fruits and vegetables for scenario S1, scenario S2 and scenario S4 (resp. Figs 5, 6 and 8). However, in scenario S3 (Fig 7), local production is insufficient to meet the population's needs for fruits and vegetables if all sugarcane parcels are preserved for the time horizon 2035. Looking at Figs 5–7, we can see that rice is the most limiting crop. This can be attributed to the limited surfaces with rice crop production potential, particularly when current agricultural practices, including sugarcane cultivation, are maintained (refer to Figs 5 and 7). Additionally, the combination of low yields, high

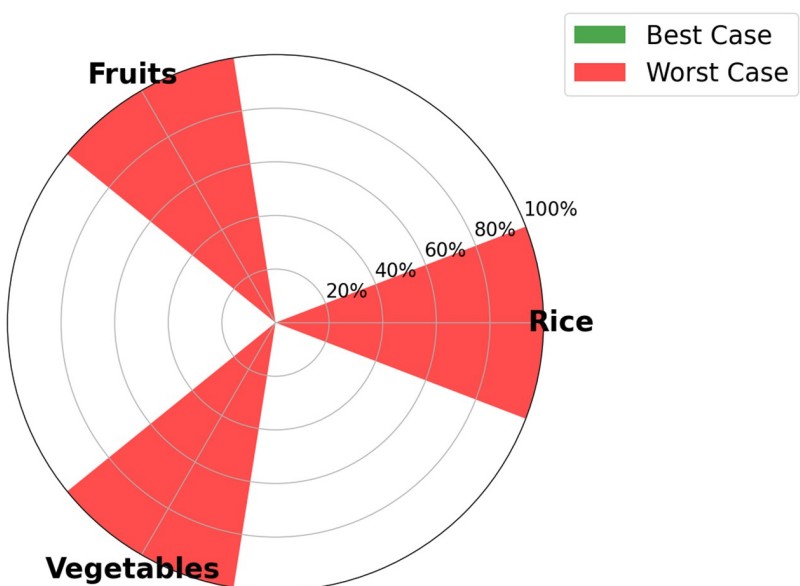

**Fig 8. Specific crop SSR for scenario S4 (Dietary + agricultural shift allowed).**

demand, and the absence of existing production on the island contributes to this limitation. Conversely, in Figs 6 and 8, the food SSR for rice crop significantly increases due to the conversion of some sugarcane parcels into rice production parcels. A 100% SSR is even reached in scenario S4 (Fig 8) as a result of lower rice consumption per household (*Mediterranean* profile). In scenario S2 (*Traditional* profile), even after converting some sugarcane parcels, complete food self sufficiency is not achieved (refer to Fig 6). Thresholds are observed on Fig 5 for scenario S1 for rice crop production, on Fig 6 for scenario S2 for rice crop production and on Fig 7 for scenario S3 for fruits, vegetables and rice crop production. In these scenarios, land use competition leads to the conversion of potential areas into crop production areas until no potential areas remain. Increasing these thresholds in our study would entail producing on areas that have been allocated no land use potentials, specifically natural protected and urban areas. Food self-consumption on existing urban parcels is not addressed here, as it does not substitute other purchases but rather leads to an additional consumption of fresh products [50]. Moreover, cultivating food on future urban parcels would result in reduced urban densification (allocated space for gardens), therefore necessitating the conversion of more agricultural areas to accommodate the rising urban growth. From these pie charts, we can deduce that scenario S4 (Dietary + agricultural shift allowed) appears to be the most resilient in relation to the food SSR. Furthermore, the potential for transitioning towards this scenario is highest in terms of food practices, given that the *Mediterranean* consumption profile currently applies to only 20% of households on the island, compared to 80% for the *Traditional* profile [50] (refer to Fig 9).

To summarize, in addition to supply-side actions (expanding agricultural areas for subsistence farming), a lever for action also exists on the demand side by shifting food consumption practices towards crops with high added-values, such as fruits and vegetables, in order to enhance the resilience of the food system. This emphasizes the need for an integrated response to be formulated by decision-makers.

**Scenarios influence on land use share.** Enhancing food system sustainability implies exploring new agricultural pathways, by assessing the impact of integrated scenarios on

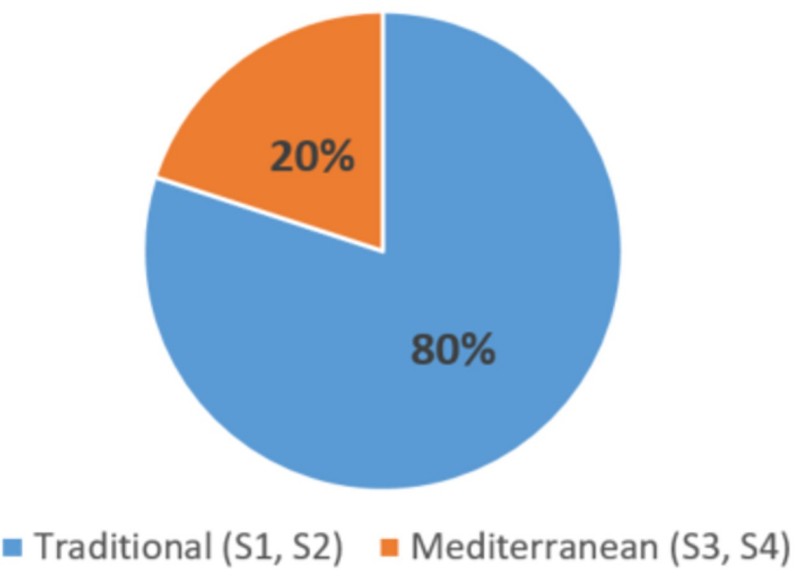

**Fig 9. Distribution of the current food consumption profiles in Reunion island.**

agricultural lands. As a result, the distribution of agricultural lands induced by the integrated scenarios and the current agricultural share are illustrated in Fig 10 for the best-case scenario in 2035. Results are presented using histograms.

First and foremost, we assume that surface areas associated with livestock and other crops remain constant for all scenarios considered. For scenarios S1 (BAU) and scenario S3 (Dietary shift), the sugarcane surface areas are maintained throughout the simulation process, constraining the development of rice, vegetables and fruits in agricultural wastelands (see specifications on constraints for land use potential allocation in Table 3). Histograms for scenario S1 and scenario S3 have the same distribution because space limitations restrict the development of agricultural production (refer to scenarios specifications in Fig 4). For scenario S2 (Agricultural shift allowed) and scenario S4 (Dietary + agricultural shift allowed), we observe a significant reduction in the share of sugarcane due to the conversion of some of these parcels into rice crop production parcels and, to a lesser extent, in vegetable and fruits production parcels. Indeed, we note a substantial portion of the UAA dedicated to rice cultivation compared to vegetables and fruits (due to low yields per hectare, high consumption and the absence of existing production on the island). Despite the conversion of some of the sugarcane areas, the sector has not been entirely abandoned, with at least 14,000 ha still under cultivation (scenario S2). Finally, we observe an increased UAA for all scenarios compared to the actual UAA, resulting from the conversion of agricultural wastelands into crop production (increase of 2,000 ha for scenario S1, S2, and S3 and 3,000 ha for scenario S4 comparing to the actual situation).

In summary, for decision-makers, regardless of the scenario under consideration, the issue is not so much the conversion of sugarcane but rather in the desire to preserve all current production. This model provides a set of agricultural pathways to be implemented to increase the resilience of the food system. Nevertheless, it means making trade-offs when selecting crops.

**Applying results on a broader scale.** Our planet can be characterized by social and ecological boundaries reflecting the level of social justice and environmental degradation. These ecological and social boundaries refer to thresholds beyond which sustainability challenges cannot be met. This conceptual framework was formalized by Kate Raworth, an economist at the University of Oxford, providing a visualization known as the "doughnut" theory [56]. This approach is best suited for small islands, which resemble miniature planets due to their defined boundaries and geographical isolation, allowing for a clear representation of empowerment

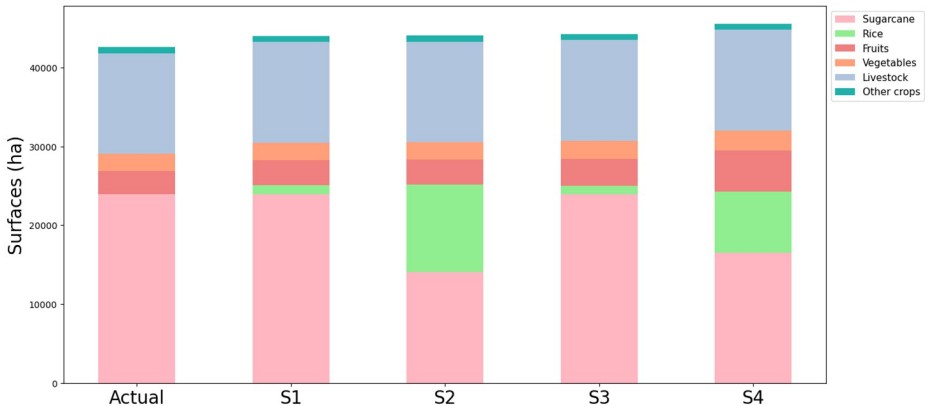

**Fig 10. UAA specified by crops according to the integrated WEF scenarios for the best case in 2035 compared to actual land use share.**

threshold effects. This approach also allows the integration of multiple interconnected resources such as water, energy, and food (the social foundation). Consequently, small islands serve as relevant 'laboratories' for investigating sustainability challenges within territories. The knowledge obtained from such studies can then be extrapolated to address broader sustainability challenges, particularly in areas such as land use management and the transition of dietary habits towards more value-added crops. The objective lies in leveraging these findings to inform policymakers at national or global levels, thereby extending the impact of the research.

## Conclusion and future work

In this article, we have presented a systemic approach to study food system sustainability in small islands thanks to the implementation of an integrated model combining a GIS and a robust optimization model. It enables the integration and the management of a broad spectrum of data, along with the definition of pertinent constraints linked to the WEF nexus. Our approach emphasizes the need to consider the nexus in terms of its impact on land use, as well as the need to measure the effects of resilience needs on land use competition. This is achieved by incorporating the food system into the WEF nexus. It facilitates the observation of thresholds, which are conditioned by land availability and induced by the resilience objective. Scenarios that integrate both supply and demand components demonstrate that changing food consumption practices, coupled with land-use policies, can effectively enhance the resilience of the food system towards sustainability. In this way, our model becomes an efficient decision support tool for local policymakers to implement informed policies on land use management that would support food system sustainability. This involves using resources with a minimal surface footprint, supporting concomitant land uses and promoting agricultural practices directed towards subsistence farming. However, effective land use management policies cannot be dissociated from reconsidering individual dietary behaviors. This potential for transitioning from the *Traditional* profile (80% of the population) to the *Mediterranean* profile (20% of the population) implies not only raising awareness of emerging deconsumption practices but also addressing the issue of the cost of living from the consumer's perspective [53] by considering that these practices are prevalent among the richest households [50]. We have focused our approach on examining thresholds and physical limits regarding the food SSR, and putting aside the economical dimension. While it can be seen as a driving aspect, our goal here was to study the land use potentials independently of such factors. The produced scenarios are insightful indicators contributing to create incentives in particular towards shifts in dietary habits, in line with the sustainability of small islands. Additional limitations of our study pertain to the granularity of spatial data collected, which needs to be aggregated at the island scale. Specifically, the use of average values, especially for agricultural yields, reflects on-the-ground reality but variations may exist across different areas. In the modeling, urbanization dynamics occur randomly on urbanizable plots, which may not necessarily capture the reality of land use planning as envisioned in local urban development plans. Also, the impacts of climate change on agricultural yields, potential PV production, or rainfall have not been evaluated. Finally, as the issue of food system sustainability in small islands is systemic, it is impossible to consider all existing interactions between resources and ecosystems. In this context, the choice of a WEF nexus approach has proven particularly relevant for addressing multiple resource management, as extensive literature is available at various geographical scales compared to other nexus approaches.

Future work includes studying the influence of uncontrolled urban growth on the food system in Reunion island, and incorporating new data from another small islands for a comparative assessment. Finally, integrating some constraints that would reflect established impacts of

climate change, as well as economical constraints relative to imports and subsidies is part of future extensions of our methodology.

## Acknowledgments

We thank all the institutions and persons who dedicated some of their precious time to participate in our interviews, as well as the reviewers for their insightful comments.

## Author Contributions

**Conceptualization:** Romain Authier, Guillaume Guimbretière, Pablo Corral-Broto.

**Data curation:** Romain Authier.

**Methodology:** Romain Authier, Benjamin Pillot, Carmen Gervet.

**Software:** Romain Authier.

**Supervision:** Carmen Gervet.

**Writing – original draft:** Romain Authier.

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
