## [Decision Letter · Decision Letter 0]

5 Apr 2024

PONE-D-24-08942Towards sustainable food systems in small islands: a Water-Energy-Food nexus approachPLOS ONE

Dear Dr. Authier,

Thank you for submitting your manuscript to PLOS ONE. After careful consideration, we feel that it has merit but does not fully meet PLOS ONE’s publication criteria as it currently stands. Therefore, we invite you to submit a revised version of the manuscript that addresses the points raised during the review process.

This study is interesting and novel. However, based on the informative comments from the reviewers, additional analysis and discussion should be provided.

We look forward to receiving your revised manuscript.

Kind regards,

Wen-Wei Sung, M.D., Ph.D.

Academic Editor

PLOS ONE

“We thank the IRD (Institut de Recherche pour le D´eveloppement) for funding the PhD work of Romain Authier, and all the institutions and persons who dedicated some of their precious time to interviews.”

“PhD grant from IRD (Institut de Recherche pour le Développement). The funders had no role in study design, data collection and analysis, decision to publish, or preparation of the manuscript.”

4. We note that Figure 3 in your submission contain [map/satellite] images which may be copyrighted. All PLOS content is published under the Creative Commons Attribution License (CC BY 4.0), which means that the manuscript, images, and Supporting Information files will be freely available online, and any third party is permitted to access, download, copy, distribute, and use these materials in any way, even commercially, with proper attribution. For these reasons, we cannot publish previously copyrighted maps or satellite images created using proprietary data, such as Google software (Google Maps, Street View, and Earth). For more information, see our copyright guidelines: http://journals.plos.org/plosone/s/licenses-and-copyright.

1. You may seek permission from the original copyright holder of Figure 3 to publish the content specifically under the CC BY 4.0 license. 

Reviewers' comments:

Reviewer's Responses to Questions

**Comments to the Author**

1. Is the manuscript technically sound, and do the data support the conclusions?

Reviewer #1: Yes

Reviewer #2: Yes

2. Has the statistical analysis been performed appropriately and rigorously? 

Reviewer #1: I Don't Know

Reviewer #2: Yes

3. Have the authors made all data underlying the findings in their manuscript fully available?

Reviewer #1: Yes

Reviewer #2: Yes

4. Is the manuscript presented in an intelligible fashion and written in standard English?

Reviewer #1: No

Reviewer #2: Yes

5. Review Comments to the Author

Reviewer #1: A very interesting study case and a fundamental issue analyzed in the ms!

Nevertheless the ms needs more work.

Main comments:

* There is no clear explanation about the optimization model. There is no mention of it in the text, only in figure 2, which says it is coded in phyton. According to table 1, the optimization model seems to be one of partial equilibrium, but it is no clear at all.

* There are no comments about model "validation" or the likelihood of model outputs. In my opinion it is a good practice that a ms like this include a section in which the authors explain why the model is reliable, and why we should be confident with its results.

* There is no discussion about the "economic aspects" of changing crops, and this is a very important issue in sustainability topics. In this context, perhaps it would be interesting to include a section that explores the "transformation effort" to increase the SSR.

* I think that some comments are needed about the relevance of this system considering the worldwide sustainability issue. Planet Earth is a very small island too, so conclusion of this research may be very interesting for addressing sustainability issues in a larger scales.

* It would be interesting to include a small comparisons of the WEF approach with other efforts, like FABLE. Why WEF is a good choice for studying sustainability issues?

Particular comments:

* Perhaps modify figure 3, in order to show where in Earth is Reunion island.

* I could not find the footnotes of figures!

* I could not find figure 10.

* The reference #35 seems to be incomplete. This is a very important reference for the ms.

* Perhaps join figures 5 to 8 into a single one, that would make easier the comparison of scenarios.

* Perhaps include a short name in the S1, S2, S3 and S4, so the text would be easier to read. For example: "S1 (traditional BAU, business as usual)".

* Perhaps include a graph like this:

+ in X axis the time (yrs);

+ in Y axis the SSR value;

+ a point showing the actual SSR value of Reunion island;

+ time trajectories of each scenario, showing their final SSR values.

This will be more easy to understand.

* Include an appendix with a more detailed explanation of interviews (how many, main results of them, que questions posed, etc.).

* Figure 9 is too simple, data do not deserve a graph. Perhaps if the 80% portion is separated in individual crops would be worthwhile to keep.

Reviewer #2: PLOS ONE

Towards sustainable food systems in small islands: a Water-Energy-Food nexus approach

Small islands face environmental and multi-sectoral challenges, which require questions of sustainability. The Water-Energy-Food (WEF) nexus is crucial for understanding the interconnections between resources and land use. A study of small island food system sustainability uses a combined GIS and optimization model to examine the reciprocal influences of local WEF resources and competition for land use. The results suggest space savings for resource development and changing food practices, particularly the conversion of sugarcane areas to subsistence agriculture.

"Introduction and related work" enough to be mentioned "Introduction"

The introduction should briefly place the study in a broad context and highlight why it is important. It must define the purpose of the work and its meaning. The current state of the research field should be carefully reviewed and key publications cited. Finally, briefly discuss the main objective of the work and highlight the main conclusions. If possible, make sure the introduction is understandable to scientists outside your particular area of research.

I have done everything possible to effectively communicate the message you wanted to convey in your introduction. The following resources can be used for this purpose.

Small islands are territories surrounded by water, with limited resources and strong economic, climatic and demographic vulnerabilities. These islands face environmental problems such as land fragility, loss of biodiversity and exposure to natural hazards and anthropogenic pressure. They are also vulnerable to climate change, energy, food and land vulnerabilities due to biophysical, socio-economic and demographic factors. Small island sustainability is a complex concept, depending on geographic, demographic and historical contexts. Resilience is a key aspect in the search for sustainability, because the lack of resilience increases the vulnerability of these fragile territories. Resilience involves improving water management and considering greater autonomy in the exploitation of local resources, such as renewable energy and local food systems. Land use is a central issue in meeting the multiple challenges of resource management and planning. The integrated Water-Energy-Food (WEF) nexus represents a promising approach to address these challenges, exploring the links between water, energy and food resources to better understand their interdependence in sustainable development.

The study focuses on the management of WEF resources on a global scale, analyzing waste management, identifying crises and identifying solutions to the decline in resource availability and accessibility. It also proposes to analyze the impacts of different agricultural production and food consumption scenarios on WEF resource production and environmental consequences using Integrated Assessment Models (IAM) and FDM for food production and Requirement. The study also proposes optimal planning of the WEF link using constrained stochastic optimization modeling using GAMS optimization software.

At the national level, the study highlights the reciprocal influences of energy prices and water resources on food prices and the role of water and energy systems on quality, availability and food accessibility. A spatio-temporal decision support model was developed for the assessment of the energy needs of the agricultural sector in Saudi Arabia.

At the local level, the methodological approaches are varied and presented as decision-making tools. For example, a multi-objective optimization model was used to evaluate WEF resource security in the Monterrey metropolitan area in Mexico, while mixed-integer nonlinear modeling was applied to the Yucheng station in China.

The study provides a decision support tool to analyze future energy consumption and greenhouse gas emissions in Tenerife, Spain, based on projections of demand for water and energy resources , as well as demographic, economic and climatic variables. Two different scenarios are proposed for 2050: a maintained trend scenario with underdeveloped renewable energies and an ecologically conscious scenario with 100% renewable energies. The approach encompasses all three dimensions of the GEF nexus and assesses sustainability for Caribbean Small Island Developing States (SIDS) with respect to the food, water and energy components. The study highlights the risks linked to the balance between supply and demand due to population growth and economic development. A user-friendly GIS-based connection platform in Taiwan, GRAT for FEW, is developed to analyze the linkages between WEF resources and trade-offs between bioenergy production, food supply, and environmental benefits. The study also offers spatio-temporal modeling of the factors of change influencing food and energy self-sufficiency through semi-structured interviews in Reunion.

This article discusses the relationship between local WEF resource use and competition for land use on small islands. It aims to address the question of food system sustainability in small islands by examining the reciprocal influences between these factors. The main challenges identified include the integration of multi-source data, the definition of heterogeneous spatio-temporal constraints, the construction of integrated WEF scenarios and the identification of food self-sufficiency thresholds. The paper presents an integrated GIS and a robust optimization methodology as a decision support tool, enabling the modeling of interactions between the WEF resource self-sufficiency process and competition for land use. The integrated GIS and optimization model aims to maximize a food self-sufficiency rate (SSR) and integrate various consumption models and agricultural sectors on the island. The document also presents a comprehensive decision support tool, integrating qualitative data from field interviews and existing literature. The article aims to answer the research question of whether the pursuit of resilience necessarily contributes to improving the sustainability of small islands within a systems approach.

"Methodology description" you should make it " Matrials and Methods"

This article focuses on food system sustainability in small island food systems, considering land use dynamics and local resource valuation within an integrated WEF approach. Population growth and economic development are considered key drivers of change within the GEF network. Urban growth impacts land use through urban sprawl and the demand for WEF resources, leading to an increased supply of these resources. The means of production implemented to meet demand determine the island's capacity to ensure the sustainability of the food system.

In the quest for resilience, dependence on imports is reduced in favor of local WEF resources. However, the development of local WEF resources can strengthen competition for land use, already reinforced by urban sprawl. This competition for land use can also limit the development of local resources due to the strong spatial constraints characteristic of small islands. Thus, there may be a maximum threshold for valuation of WEF resources due to competition for land use.

The integrated model must fulfill several objectives: (1) accounting for competition for land use in the context of the use of local WEF resources and (2) identifying thresholds of the food self-sufficiency process at- beyond which it affects the sustainability of the food system. The model takes into account various layers of geographic data and generates a land use potential map, which is then converted into a set of parcel attribute lists for the land use optimization model.

Three distinct land use potentials are defined: (i) agricultural production potential, (ii) urban growth potential and (iii) electricity production potential. The nexus approach developed here implies that some parcels have the potential to move from their current land use to one or more land uses, taking into account concomitant land uses.

The Land Use Potential Allocation Model is a tool used to assign potential land use to parcels based on their land use potential. The model adheres to predetermined constraints, such as minimum area, maximum elevation, maximum slope, and specific land use criteria for urban growth potential, electricity generation potential, and production potential. agricultural. The model also takes into account water requirements for agricultural production potential.

The model generates a map of land use potentials, which is converted into lists of parcel attributes for the optimization model. The model systematically identifies the optimal land use or combination of land uses for each parcel from its range of potential land uses, allowing for concurrent land uses on a single parcel. The model is run under the constraints associated with the GEF link and incorporates an objective function for the 2035 time horizon, aiming to maximize food sustainability.

The optimization problem is defined using multiple optimization constraints that integrate factors influencing change within the WEF network, with a specific focus on resilience to improve sustainability. The constraints include food production constraints, electricity production constraints, energy import constraints, intermittent renewable energy production constraints, solar self-consumption constraints, urban sprawl constraints and land use conversion of potential areas.

This case study explores issues related to the sustainability of the food system in Réunion, a French overseas department located in the Indian Ocean. The island is heavily dependent on imported food and energy resources and is vulnerable to disruptions that could interrupt its external supplies. It faces land pressure linked to urban sprawl. The French government has expressed its desire to achieve food and energy self-sufficiency by 2030.

The main data used in this study are geographic data layers, including land use layer, cadastral parcel layer, topography layer, irrigated perimeter layer, practical photovoltaic energy potential layer and the precipitation layer. The WEF Nexus approach is implemented in Réunion, taking into account various energy sources, including imported biomass, oil, coal, wind, photovoltaics, hydropower and local biomass.

The study also assesses the direct land use impact of WEF resources and urban growth to observe the dynamics of land use change and land use competition. The direct impact on land use linked to biomass from agricultural residues (in this case sugarcane bagasse) is taken into account, while the direct impact on land use of imported resources (coal, oil, imported biomass) is zero.

The direct future impact on land use associated with rice, fruits, vegetables, water and urban growth is considered a consequence of the island's population growth. Building integrated GEF scenarios, combining demand-side and supply-side approaches, through online literature and field interviews, allows for a more complete description of the link.

For four months, field interviews were conducted with different local actors in Reunion Island to contribute to the development of the scenarios. The interviews revealed a growing awareness of the island's vulnerabilities, including its dependence on food imports. Associations like the Réunion Rice Association and the P´ei Rice Growers aim to reintroduce rice cultivation on the island, while citizen groups like Oasis Réunion position themselves as a force for proposals for complete food self-sufficiency. The cultivation of sugar cane is debated, with some seeing it as an obstacle to food self-sufficiency, while others plead for its preservation.

Scenarios were developed to describe the diversity of visions of local stakeholders and propose compromises to facilitate decision-making. The scenarios included demand-side water needs, energy demand, food demand, and supply-side water needs. Two food consumption profiles were selected: Traditional and Mediterranean.

Energy demand was estimated using existing consumption data from 2022, with future increases dependent on population growth and the expansion of electric vehicle fleets. Food consumption profiles were oriented towards local production when food products were produced locally.

The scenarios were designed by combining different food consumption profiles (demand side) with agricultural practices (supply side) to create four distinct integrated scenarios. Constraints for each land use potential are summarized in Tables S1, S2, S3 and S4. The optimization model was used to analyze the data related to the static parameters of the optimization model.

The study aims to improve the sustainability of the food system by 2035 by identifying the thresholds of the food self-sufficiency process and evaluating the share of land use. The electricity mix can influence food self-sufficiency, particularly when energy projects with significant impacts on land use, such as ground-mounted photovoltaic installations, are developed. The study selects an electricity mix relying on ground-mounted photovoltaic panels with limited biomass imports, which has a less favorable impact on food self-sufficiency due to increased land use competition, but mitigates dependence on imported energy resources.

The optimization model visualizes the maximum food self-sufficiency for each crop by 2035, with the donut chart highlighting both the best and worst case scenarios. Food self-sufficiency in fruits and vegetables reaches 100% for scenarios S1, S2 and S4. Rice is the most limiting crop due to its limited production potential, low yields, high demand and no existing production on the island.

The S4 scenario appears to be the most resilient in terms of food self-sufficiency, with the potential for transition to this scenario being the highest in terms of dietary practices. In addition to supply-driven actions, an integrated response is needed by reorienting food consumption practices towards high-value crops, such as fruits and vegetables, to strengthen the resilience of the food system.

The study explores the impact of integrated scenarios on agricultural land to improve the sustainability of the food system. The distribution of agricultural land and the current agricultural share are illustrated in Figure 10 for the best scenario in 2035. The study assumes that the areas associated with livestock and other crops remain constant. However, for scenarios S1 and S3, sugarcane areas are maintained, limiting the development of rice, vegetables and fruits in agricultural wastelands. For scenarios S2 and S4, significant reductions in the share of sugarcane are observed due to conversions to rice production plots and agricultural wastelands. The model proposes a set of agricultural pathways to increase food system resilience, but requires making trade-offs when selecting crops.

Authors should discuss the results and how they can be interpreted from the perspective of previous studies and working hypotheses. The results and their implications should be discussed in the broadest possible context. Future research directions may also be highlighted.

Conclusion and future work

This paper presents a systems approach to study food system sustainability in small islands using a GIS model and a robust optimization model. The model incorporates data and constraints related to the WEF nexus, focusing on land use and resilience needs. It makes it possible to identify thresholds conditioned by land availability and resilience objectives. The model is an effective decision-making tool for local policy makers to implement land use management policies for food system sustainability. It highlights the importance of taking into account individual dietary behaviors and the cost of living during the transition from traditional populations to Mediterranean populations.

the authors should briefly mention the limitation of them study

6. PLOS authors have the option to publish the peer review history of their article (what does this mean?). If published, this will include your full peer review and any attached files.

Reviewer #1: **Yes: **Ernesto Vicente Vega Peña

Reviewer #2: **Yes: **Mohamed A. E. AbdelRahman

---

## [Author Response · Author response to Decision Letter 0]

21 May 2024

Thanks to the reviewers and to the editor for their insightful comments. The point-by-point responses to their comments have been added to the ‘Response to reviewers’ letter.

---

## [Decision Letter · Decision Letter 1]

25 Jun 2024

PONE-D-24-08942R1Towards sustainable food systems in small islands: a Water-Energy-Food nexus approachPLOS ONE

Dear Dr. Authier,

Thank you for submitting your manuscript to PLOS ONE. After careful consideration, we feel that it has merit but does not fully meet PLOS ONE’s publication criteria as it currently stands. Therefore, we invite you to submit a revised version of the manuscript that addresses the points raised during the review process.

Please address the reviewer's comments to complete the minor revision.

We look forward to receiving your revised manuscript.

Kind regards,

Wen-Wei Sung, M.D., Ph.D.

Academic Editor

PLOS ONE

Journal Requirements:

Reviewers' comments:

Reviewer's Responses to Questions

**Comments to the Author**

1. If the authors have adequately addressed your comments raised in a previous round of review and you feel that this manuscript is now acceptable for publication, you may indicate that here to bypass the “Comments to the Author” section, enter your conflict of interest statement in the “Confidential to Editor” section, and submit your "Accept" recommendation.

Reviewer #1: All comments have been addressed

Reviewer #2: All comments have been addressed

2. Is the manuscript technically sound, and do the data support the conclusions?

Reviewer #1: Yes

Reviewer #2: No

3. Has the statistical analysis been performed appropriately and rigorously? 

Reviewer #1: Yes

Reviewer #2: No

4. Have the authors made all data underlying the findings in their manuscript fully available?

Reviewer #1: Yes

Reviewer #2: Yes

5. Is the manuscript presented in an intelligible fashion and written in standard English?

Reviewer #1: Yes

Reviewer #2: Yes

6. Review Comments to the Author

Reviewer #1: All comments were attended by the authors. The issue of model validation was not addressed as I would expect. Confidence intervals of model outputs may not be enough, as they are not useful to understand theoretical reliability of the model.

Reviewer #2: . The knowledge obtained from such studies can then be extrapolated to address broader sustainability challenges, particularly in areas such as land use management and the transition of dietary habits towards more value-added crops. The objective lies in leveraging these findings to inform policymakers at national or global levels, thereby extending the impact of the research. (THE LINES (636636–639) YOU SHOULD refer TO

1- DOI: 10.1080/27669645.2022.2103953. To link to this article: https://doi.org/10.1080/27669645.2022.2103953

2- https://doi.org/10.1007/s12210-023-01155-3

Important comments: why you didn't implement my efforts in the previous round?!

in my previous review round, itried to:

Considering the significance of this work in addressing a crucial topic for future agricultural planning, I decided to approach it differently. Instead of providing negative feedback that the authors may not be able to incorporate, I empathized with their perspective and made earnest efforts to draft these paragraphs. My intention is for the authors to appreciate and potentially incorporate these suggestions into their valuable manuscript. I am confident that this approach will enhance the presentation of the manuscript and effectively convey a direct and coherent message to the readers.

Kindly note while I am trying to put some efforts for your work, I considered the followings:

1-Abstract: should give a pertinent overview of the work. We strongly encourage authors to use the following style of structured abstracts, but without headings: (1) Background: Place the question addressed in a broad context and highlight the purpose of the study; (2) Methods: briefly describe the main methods or treatments applied; (3) Results: summarize the article's main findings; (4) Conclusions: indicate the main conclusions or interpretations. The abstract should be an objective representation of the ar-ticle and it must not contain results that are not presented and substantiated in the main text and should not exaggerate the main conclusions.

2- Introduction

The introduction should briefly place the study in a broad context and highlight why it is important. It should define the purpose of the work and its significance. The current state of the research field should be carefully reviewed and key publications cited. Please highlight controversial and diverging hypotheses when necessary. Finally, briefly men-tion the main aim of the work and highlight the principal conclusions.

But for Discussion

Authors should discuss the results and how they can be interpreted from the per-spective of previous studies and of the working hypotheses. The findings and their implications should be discussed in the broadest context possible. Future research directions may also be highlighted.

my previous efforts was as follows:

PLOS ONE

Towards sustainable food systems in small islands: a Water-Energy-Food nexus approach

Small islands face environmental and multi-sectoral challenges, which require questions of sustainability. The Water-Energy-Food (WEF) nexus is crucial for understanding the interconnections between resources and land use. A study of small island food system sustainability uses a combined GIS and optimization model to examine the reciprocal influences of local WEF resources and competition for land use. The results suggest space savings for resource development and changing food practices, particularly the conversion of sugarcane areas to subsistence agriculture.

"Introduction and related work" enough to be mentioned "Introduction"

The introduction should briefly place the study in a broad context and highlight why it is important. It must define the purpose of the work and its meaning. The current state of the research field should be carefully reviewed and key publications cited. Finally, briefly discuss the main objective of the work and highlight the main conclusions. If possible, make sure the introduction is understandable to scientists outside your particular area of research.

I have done everything possible to effectively communicate the message you wanted to convey in your introduction. The following resources can be used for this purpose.

Small islands are territories surrounded by water, with limited resources and strong economic, climatic and demographic vulnerabilities. These islands face environmental problems such as land fragility, loss of biodiversity and exposure to natural hazards and anthropogenic pressure. They are also vulnerable to climate change, energy, food and land vulnerabilities due to biophysical, socio-economic and demographic factors. Small island sustainability is a complex concept, depending on geographic, demographic and historical contexts. Resilience is a key aspect in the search for sustainability, because the lack of resilience increases the vulnerability of these fragile territories. Resilience involves improving water management and considering greater autonomy in the exploitation of local resources, such as renewable energy and local food systems. Land use is a central issue in meeting the multiple challenges of resource management and planning. The integrated Water-Energy-Food (WEF) nexus represents a promising approach to address these challenges, exploring the links between water, energy and food resources to better understand their interdependence in sustainable development.

The study focuses on the management of WEF resources on a global scale, analyzing waste management, identifying crises and identifying solutions to the decline in resource availability and accessibility. It also proposes to analyze the impacts of different agricultural production and food consumption scenarios on WEF resource production and environmental consequences using Integrated Assessment Models (IAM) and FDM for food production and Requirement. The study also proposes optimal planning of the WEF link using constrained stochastic optimization modeling using GAMS optimization software.

At the national level, the study highlights the reciprocal influences of energy prices and water resources on food prices and the role of water and energy systems on quality, availability and food accessibility. A spatio-temporal decision support model was developed for the assessment of the energy needs of the agricultural sector in Saudi Arabia.

At the local level, the methodological approaches are varied and presented as decision-making tools. For example, a multi-objective optimization model was used to evaluate WEF resource security in the Monterrey metropolitan area in Mexico, while mixed-integer nonlinear modeling was applied to the Yucheng station in China.

The study provides a decision support tool to analyze future energy consumption and greenhouse gas emissions in Tenerife, Spain, based on projections of demand for water and energy resources , as well as demographic, economic and climatic variables. Two different scenarios are proposed for 2050: a maintained trend scenario with underdeveloped renewable energies and an ecologically conscious scenario with 100% renewable energies. The approach encompasses all three dimensions of the GEF nexus and assesses sustainability for Caribbean Small Island Developing States (SIDS) with respect to the food, water and energy components. The study highlights the risks linked to the balance between supply and demand due to population growth and economic development. A user-friendly GIS-based connection platform in Taiwan, GRAT for FEW, is developed to analyze the linkages between WEF resources and trade-offs between bioenergy production, food supply, and environmental benefits. The study also offers spatio-temporal modeling of the factors of change influencing food and energy self-sufficiency through semi-structured interviews in Reunion.

This article discusses the relationship between local WEF resource use and competition for land use on small islands. It aims to address the question of food system sustainability in small islands by examining the reciprocal influences between these factors. The main challenges identified include the integration of multi-source data, the definition of heterogeneous spatio-temporal constraints, the construction of integrated WEF scenarios and the identification of food self-sufficiency thresholds. The paper presents an integrated GIS and a robust optimization methodology as a decision support tool, enabling the modeling of interactions between the WEF resource self-sufficiency process and competition for land use. The integrated GIS and optimization model aims to maximize a food self-sufficiency rate (SSR) and integrate various consumption models and agricultural sectors on the island. The document also presents a comprehensive decision support tool, integrating qualitative data from field interviews and existing literature. The article aims to answer the research question of whether the pursuit of resilience necessarily contributes to improving the sustainability of small islands within a systems approach.

"Methodology description" you should make it " Matrials and Methods"

This article focuses on food system sustainability in small island food systems, considering land use dynamics and local resource valuation within an integrated WEF approach. Population growth and economic development are considered key drivers of change within the GEF network. Urban growth impacts land use through urban sprawl and the demand for WEF resources, leading to an increased supply of these resources. The means of production implemented to meet demand determine the island's capacity to ensure the sustainability of the food system.

In the quest for resilience, dependence on imports is reduced in favor of local WEF resources. However, the development of local WEF resources can strengthen competition for land use, already reinforced by urban sprawl. This competition for land use can also limit the development of local resources due to the strong spatial constraints characteristic of small islands. Thus, there may be a maximum threshold for valuation of WEF resources due to competition for land use.

The integrated model must fulfill several objectives: (1) accounting for competition for land use in the context of the use of local WEF resources and (2) identifying thresholds of the food self-sufficiency process at- beyond which it affects the sustainability of the food system. The model takes into account various layers of geographic data and generates a land use potential map, which is then converted into a set of parcel attribute lists for the land use optimization model.

Three distinct land use potentials are defined: (i) agricultural production potential, (ii) urban growth potential and (iii) electricity production potential. The nexus approach developed here implies that some parcels have the potential to move from their current land use to one or more land uses, taking into account concomitant land uses.

The Land Use Potential Allocation Model is a tool used to assign potential land use to parcels based on their land use potential. The model adheres to predetermined constraints, such as minimum area, maximum elevation, maximum slope, and specific land use criteria for urban growth potential, electricity generation potential, and production potential. agricultural. The model also takes into account water requirements for agricultural production potential.

The model generates a map of land use potentials, which is converted into lists of parcel attributes for the optimization model. The model systematically identifies the optimal land use or combination of land uses for each parcel from its range of potential land uses, allowing for concurrent land uses on a single parcel. The model is run under the constraints associated with the GEF link and incorporates an objective function for the 2035 time horizon, aiming to maximize food sustainability.

The optimization problem is defined using multiple optimization constraints that integrate factors influencing change within the WEF network, with a specific focus on resilience to improve sustainability. The constraints include food production constraints, electricity production constraints, energy import constraints, intermittent renewable energy production constraints, solar self-consumption constraints, urban sprawl constraints and land use conversion of potential areas.

This case study explores issues related to the sustainability of the food system in Réunion, a French overseas department located in the Indian Ocean. The island is heavily dependent on imported food and energy resources and is vulnerable to disruptions that could interrupt its external supplies. It faces land pressure linked to urban sprawl. The French government has expressed its desire to achieve food and energy self-sufficiency by 2030.

The main data used in this study are geographic data layers, including land use layer, cadastral parcel layer, topography layer, irrigated perimeter layer, practical photovoltaic energy potential layer and the precipitation layer. The WEF Nexus approach is implemented in Réunion, taking into account various energy sources, including imported biomass, oil, coal, wind, photovoltaics, hydropower and local biomass.

The study also assesses the direct land use impact of WEF resources and urban growth to observe the dynamics of land use change and land use competition. The direct impact on land use linked to biomass from agricultural residues (in this case sugarcane bagasse) is taken into account, while the direct impact on land use of imported resources (coal, oil, imported biomass) is zero.

The direct future impact on land use associated with rice, fruits, vegetables, water and urban growth is considered a consequence of the island's population growth. Building integrated GEF scenarios, combining demand-side and supply-side approaches, through online literature and field interviews, allows for a more complete description of the link.

For four months, field interviews were conducted with different local actors in Reunion Island to contribute to the development of the scenarios. The interviews revealed a growing awareness of the island's vulnerabilities, including its dependence on food imports. Associations like the Réunion Rice Association and the P´ei Rice Growers aim to reintroduce rice cultivation on the island, while citizen groups like Oasis Réunion position themselves as a force for proposals for complete food self-sufficiency. The cultivation of sugar cane is debated, with some seeing it as an obstacle to food self-sufficiency, while others plead for its preservation.

Scenarios were developed to describe the diversity of visions of local stakeholders and propose compromises to facilitate decision-making. The scenarios included demand-side water needs, energy demand, food demand, and supply-side water needs. Two food consumption profiles were selected: Traditional and Mediterranean.

Energy demand was estimated using existing consumption data from 2022, with future increases dependent on population growth and the expansion of electric vehicle fleets. Food consumption profiles were oriented towards local production when food products were produced locally.

The scenarios were designed by combining different food consumption profiles (demand side) with agricultural practices (supply side) to create four distinct integrated scenarios. Constraints for each land use potential are summarized in Tables S1, S2, S3 and S4. The optimization model was used to analyze the data related to the static parameters of the optimization model.

The study aims to improve the sustainability of the food system by 2035 by identifying the thresholds of the food self-sufficiency process and evaluating the share of land use. The electricity mix can influence food self-sufficiency, particularly when energy projects with significant impacts on land use, such as ground-mounted photovoltaic installations, are developed. The study selects an electricity mix relying on ground-mounted photovoltaic panels with limited biomass imports, which has a less favorable impact on food self-sufficiency due to increased land use competition, but mitigates dependence on imported energy resources.

The optimization model visualizes the maximum food self-sufficiency for each crop by 2035, with the donut chart highlighting both the best and worst case scenarios. Food self-sufficiency in fruits and vegetables reaches 100% for scenarios S1, S2 and S4. Rice is the most limiting crop due to its limited production potential, low yields, high demand and no existing production on the island.

The S4 scenario appears to be the most resilient in terms of food self-sufficiency, with the potential for transition to this scenario being the highest in terms of dietary practices. In addition to supply-driven actions, an integrated response is needed by reorienting food consumption practices towards high-value crops, such as fruits and vegetables, to strengthen the resilience of the food system.

The study explores the impact of integrated scenarios on agricultural land to improve the sustainability of the food system. The distribution of agricultural land and the current agricultural share are illustrated in Figure 10 for the best scenario in 2035. The study assumes that the areas associated with livestock and other crops remain constant. However, for scenarios S1 and S3, sugarcane areas are maintained, limiting the development of rice, vegetables and fruits in agricultural wastelands. For scenarios S2 and S4, significant reductions in the share of sugarcane are observed due to conversions to rice production plots and agricultural wastelands. The model proposes a set of agricultural pathways to increase food system resilience, but requires making trade-offs when selecting crops.

Authors should discuss the results and how they can be interpreted from the perspective of previous studies and working hypotheses. The results and their implications should be discussed in the broadest possible context. Future research directions may also be highlighted.

Conclusion and future work

This paper presents a systems approach to study food system sustainability in small islands using a GIS model and a robust optimization model. The model incorporates data and constraints related to the WEF nexus, focusing on land use and resilience needs. It makes it possible to identify thresholds conditioned by land availability and resilience objectives. The model is an effective decision-making tool for local policy makers to implement land use management policies for food system sustainability. It highlights the importance of taking into account individual dietary behaviors and the cost of living during the transition from traditional populations to Mediterranean populations.

the authors should briefly mention the limitation of them study

7. PLOS authors have the option to publish the peer review history of their article (what does this mean?). If published, this will include your full peer review and any attached files.

Reviewer #1: **Yes: **Ernesto Vicente Vega Peña

Reviewer #2: **Yes: **Mohamed A. E. AbdelRahman

---

## [Author Response · Author response to Decision Letter 1]

22 Aug 2024

The reviewers' comments have been addressed in the response letter to the reviewers

---

## [Editor Report · Decision Letter 2]

5 Sep 2024

Towards sustainable land management in small islands : a Water-Energy-Food nexus approach

PONE-D-24-08942R2

Dear Dr. Authier,

We’re pleased to inform you that your manuscript has been judged scientifically suitable for publication and will be formally accepted for publication once it meets all outstanding technical requirements.

Kind regards,

Wen-Wei Sung, M.D., Ph.D.

Academic Editor

PLOS ONE
---

## [Editor Report · Acceptance letter]

11 Oct 2024

PONE-D-24-08942R2 

PLOS ONE

Dear Dr. Authier, 

I'm pleased to inform you that your manuscript has been deemed suitable for publication in PLOS ONE. Congratulations! Your manuscript is now being handed over to our production team.

Kind regards, 

on behalf of

Dr. Wen-Wei Sung 

Academic Editor

PLOS ONE